# Not Only Immune Escape—The Confusing Role of the TRP Metabolic Pathway in Carcinogenesis

**DOI:** 10.3390/cancers13112667

**Published:** 2021-05-28

**Authors:** Iwona Kwiatkowska, Justyna Magdalena Hermanowicz, Alicja Przybyszewska-Podstawka, Dariusz Pawlak

**Affiliations:** 1Department of Pharmacodynamics, Medical University of Bialystok, Mickiewicza 2C, 15-222 Bialystok, Poland; justyna.hermanowicz@umb.edu.pl (J.M.H.); dariusz.pawlak@umb.edu.pl (D.P.); 2Department of Clinical Pharmacy, Medical University of Bialystok, Mickiewicza 2C, 15-222 Bialystok, Poland; 3Department of Biochemistry and Molecular Biology, Medical University of Lublin, Chodźki 1, 20-093 Lublin, Poland; alicjaprzybyszewska@umlub.pl

**Keywords:** kynurenine pathway, tryptophan, epithelial–mesenchymal transition, carcinogenesis, circulating tumor cells

## Abstract

**Simple Summary:**

Recently, mechanisms that lead to immune escape by cancer cells have been under great investigation. Elements involved in the tryptophan metabolism pathway and its derivatives are considered factors that play a wide role in silencing the immune system. However, it seems that those agents contribute to tumorigenesis through a direct impact on cancer cells. This study aimed to gather available data about the kynurenine pathway and its modulating effects on disease development through the impact on immune and cancer cells. This allows for an understanding of the complexity of this metabolic pathway in the context of carcinogenesis and indicates ambiguities that may explain the current failure of therapy with the use of compounds inhibiting tryptophan metabolism. The collected data not only help us to understand the pathogenesis of cancer but also provide the basis for the development of new therapeutic strategies in oncology.

**Abstract:**

Background: The recently discovered phenomenon that cancer cells can avoid immune response has gained scientists’ interest. One of the pathways involved in this process is tryptophan (TRP) metabolism through the kynurenine pathway (KP). Individual components involved in TRP conversion seem to contribute to cancerogenesis both through a direct impact on cancer cells and the modulation of immune cell functionality. Due to this fact, this pathway may serve as a target for immunotherapy and attempts are being made to create novel compounds effective in cancer treatment. However, the results obtained from clinical trials are not satisfactory, which raises questions about the exact role of KP elements in tumorigenesis. An increasing number of experiments reveal that TRP metabolites may either be tumor promoters and suppressors and this is why further research in this field is highly needed. The aim of this study is to present KP as a modulator of cancer development through multiple mechanisms and to point to its ambiguity, which may be a reason for failures in treatment based on the inhibition of tryptophan metabolism

## 1. Introduction

The intensive development of science in the field of immuno-oncology gives hope for a thorough understanding of the changes taking place in the body during the development of neoplasms. One of the pathways that has been particularly strongly studied in this context is the metabolism of tryptophan (TRP). Efforts are being made to assign a pro- or anti-tumor role to individual elements of this metabolic pathway. However, the complexity and multiplicity of intertwined processes still leave more questions than answers and represent an open field for future research. This publication summarizes the available data on the direct and indirect effects of components of the TRP metabolic pathway on carcinogenesis.

## 2. Mechanisms Involved in Immune Evasion

For many years, the role of the immune system in cancer development has been under detailed scientific investigation. It is known that immune cells have an ability to recognize developing malignant cells. On the other hand, the enhanced activity of regulatory Tcells (Treg) leads to a decreased immune response and facilitates tumor growth. Moreover, cancer cells secrete factors that favor immunosuppressive microenvironment development, which further enables them to avoid the immune mechanism. This phenomenon is called “immunoediting” and is based on the hypothesis, that the immune system can be both—a tumor suppressor and promotor. It is divided into three phases—elimination, equilibrium, and escape. The first stage is based on observations that cancer cells express specific antigens that are recognized by dendritic cells (DCs) and presented to lymphocytes T, which in turn eliminate pathological cells through cytotoxic mechanisms [1]. In this step, tumor cells can be recognized also by macrophages and natural killers, which altogether leads to Tcell activation [2]. If the immune system eliminates all of the abnormal cells, the whole process of tumor development stops. However, some growing cells seem to be resistant to host immune mechanisms, which leads to the second phase—equilibrium. In this step, dynamic processes remain in balance, but cancer cells gain features that allow them to avoid immune recognition with the following destruction. Factors such as IL-12, IL-23 (interleukin 12, 23) and the elements of adaptive immune response keep tumor cells in a silent state [3]. However, if malignant cells obtain superiority, they progress to the third phase of immunoediting—escape—and become clinically apparent. To reach this state, cancer cells expand multiple mechanisms which facilitate their immunological evasion. One of them is based on a decreased expression of MHC-I (major histocompatibility complex, class I), with the following disturbances in antigen expression or presentation. This leads to inhibition in an antigenicity and moderates a recognition of tumor cells by immune factors. Another way to silence host defense response toward developing malignant cells is the production of molecules, which serve as immune inhibitors. Among them, programmed death-ligand-1 (PD-L1) is one of the most studied and well-known molecules, which already serves as a pharmacological target point in clinical practice. This ligand, after binding to its receptor PD-1 (programmed cell death protein 1) on immune cells, exerts multiple effects on them. Enhanced conversion of CD4+ Tcells into immunosuppressive Treg, decreased cytotoxicity of CD8+ Tcells, reprogramming macrophages into M2 subtypes, which inhibit immunity, are among well-studied effects acquired after PD-L1/PD-1 pathway activation [4,5]. The third and the most complex mechanism triggered by cancer cells for immune evasion is the excretion of prosurvival factors and molecules, which enable the production of an immunosuppressive microenvironment. It can be obtained by the secretion of cytokines with the following recruitment of Treg and MDSCs (myeloid-derived suppressor cells), and changes in amino-acids metabolism. Arginine, glutamine, leucine, and tryptophan are among those which are now under investigation and establishment of their function will help to better understand the biology of cancer [6,7,8]. Due to the broad role of each of the listed amino-acids, it is not possible to describe them in detail in one manuscript. For this reason, the given paper focuses selectively on one of them—TRP—and its metabolism via the kynurenine pathway (KP) as a factor enhancing tumor development.

## 3. Tryptophan Metabolism and Its Modulators

Tryptophan, an endogenous amino-acid essential for proper organism development in the course of further metabolic transformations, is converted into indole, with the participation of intestinal microflora, into serotonin (5-HT) under the influence of TRP hydroxylase 2 enzymes, and in the highest level is metabolized through kynurenine pathway [9]. The latter involved two isoforms of indoleamine oxidases (IDO1, IDO2) and tryptophan 2,3-oxidase (TDO2) which are rate-limiting enzymes, that degrade TRP into kynurenine (Kyn). The next steps lead to the transformation of Kyn into kynurenic acid (KYNA), 3-hydroxykynurenine (3-HKYN), antranilic acid (AA) and further production of xanturenic (XA), picolinic (PA) and quinolinic (QUIN) acids. In the final step, active NAD^+^ (nicotinamide adenine dinucleotide) arises. Most TRP metabolites are active and exert a multiple and differentiated role in cancer development, which is described in detail below. Here, it should be emphasized that malignant cells can produce individual elements of KP, as well as factors that enhance the activity of this pathway. IDO1 expression is observed in almost all human tissues and its expression upraises with age, IDO2 at the highest level can be observed in the liver, epididymis and brain, and TDO2 except the liver and brain, can be found in the placenta [10]. All of them may be detected in different types of cancers with various severity. Those three enzymes, as rate-limiting factors of TRP metabolism, gained the greatest scientific interest and are now under intense development as a potential therapeutic goal in cancer immunotherapy. Understanding which factors take part in controlling KP elements can shed a light on new therapeutic strategies in oncology. The main molecules involved in IDO-1 activity regulation are proinflammatory agents, i.e., lipopolysaccharides, pathogen-associated molecular patterns, TGF-β (transforming growth factor-beta), and IFN-γ (interferon-gamma) at the forefront. Chronic inflammation is a hallmark of cancer, so the involvement of the aforementioned factors in both processes, i.e., inflammation and expression of TRP catabolizing enzymes, suggests that kynurenine pathway elements play a role in carcinogenesis. An additional argument indicating the involvement of KP in the development of tumorigenesis is an observed correlation between the presence of the known proto-oncogene MYC and overexpression of tryptophan transporters and its increased intracellular transport. Furthermore, the level of KP enzymes is significantly higher in the presence of MYC, when compared to a knock-out cell line [11]. Another oncogene, whose activity influences IDO1 regulation, is the c-KIT proto-oncogene. Balachandran et al. showed that the inhibition of KIT signaling significantly decreases IDO1 activity, confirming the role of this oncogene in KP regulation [12]. Conversely, the high activity of tumor suppressor protein Bin-1 was connected with low IDO1 expression and better prognosis for patients [13], which additionally links the TRP metabolite pathway with tumor development.

An expression of TDO2 is controlled by corticosteroids, and it was shown that cortisol increases gene transcription of this protein [14]. Other hormones, such as estrogen and testosterone, seem to have no impact on TDO expression [15,16]. Still, TRP itself induces TDO2 expression and thus its high dietary intake is a stimulator for this protein. Moreover, in mice models, a high-fat diet was indicated as a factor uprising liver TDO2 level [17]. This may point to a potential role of a balanced diet in cancer prevention. In regards to cancer development and KP regulation, the role of the active form of vitamin B6 (PLP) cannot be overlooked, as its low serum level is correlated with a higher risk of disease occurrence. PLP is a cofactor of KP enzymes, i.e., kynureninase and kynurenine aminotraspherases, which take part in the transformation of Kyn to AA and HK to HAA. Therefore, low B6 supply and its systemic deficiency lead to the accumulation of procancerous metabolites [18]. The gathered information shows how complex and multifactorial the tryptophan metabolism pathway is. Moreover, most of the formed metabolites enhance cancerogenesis, but some of them seem to play a protective role, which makes the whole pathway more challenging to use as a therapeutic target. The further part of this manuscript describes the known aspects of KP elements in cancerogenesis modulation (Figure 1).

## 4. IDO1 and Its Role in Cancer Development

IDO1, the best-studied enzyme of all KP, seems to play one of the most complex roles in cancerogenesis. TRP depletion in local milieu IDO1 affects immunological cells and silences immune response. The activation of GCN-2 kinase (general control nonderepressible 2 kinase) and the inhibition of mTOR (mechanistic target of rapamycin) pathway lead to an immunosuppressive phenotype and facilitated immunosurveillance escape by cancer cells. It has been discovered that IDO1 has a direct impact on malignant cells and enhances tumorigenesis by increasing angiogenesis and metastasis. Moreover, this enzyme conditions metabolite production which contributes to oncogenesis either through AhR (aryl hydrocarbon receptor) activation or in other mechanisms described below. Due to that fact, IDO1, as a pharmacological target, has a high potential in oncology and its inhibition gives an opportunity for increased therapeutic successes. The depletion of TRP affects dendritic cells and two populations of lymphocyte T—immunosuppressive regulatory Tcells (Treg) and cytotoxic lymphocyte T (Tc). TRP deficiency, through the activation of GNC-2, is a signal enhancing the expression of inhibitory receptors (ILT3, ILT4) in DCs [19]. Those cells, as antigen-presenting cells (APCs) to lymphocytes T, as well take part in the differentiation of naïve Tcells in chosen subpopulations. The overexpression of ILTs, especially ILT3 and ILT4 on DCs, induces the differentiation of CD8+ and CD4+ Tcells into Treg [20]. Additionally, in CD4+ Th cells, the presence of ILTs enhances their anergy, thus inhibits the antitumor role. A high expression of ILTs essential for Treg induction [21] altogether favors the immunosuppressive phenotype. In general, those receptors are suspected to be a key factor inducing tolerogenicity [22] thus developing immunological tolerance in host organisms. The role of immunoglobulin-like transcript receptors in cancer development has been reported previously [23]. In few cancer types, such as breast, colorectal, non-small cell lung cancer, and renal cell carcinoma, high expression of those proteins was generally connected with more advanced stages of the disease, more often occurring metastasis, and poor prognosis for patients. The mentioned research points on the activation of ERK1/2 (extracellular signal-regulated kinases) signaling pathway, enhanced epithelial-to-mesenchymal transition (EMT), and increase in VEGF (vascular endothelial growth factor) level, which altogether contributes to augmented motility, angiogenesis, and invasiveness of cancer cells [24,25,26,27]. The gathered information shows that TRP depletion affects immune cells, which has a direct impact on cancer cells, their increased motility, metastasis potential and results in patients’ worse prognosis and overall survival. Another structure affected by GCN-2 activation exerted by IDO-1-induced TRP depletion is ζ-chain of T-cell receptor (TCR) in CD8+ Tcells. This structure is a key agent which conditions the occurrence of signaling from TCR [28], with the following activation of CD8+ and their full antitumor immune response [29]. Therefore, its downregulation in case of non-sufficient TRP level interrupts the cascade of events with the following impairment in lymphocyte proliferation and cytokine production [30]. To fully understand how the downregulation of the zeta chain in TCR affects tumor development, it is necessary to describe the role of CD8+ lymphocytes in this process. Those cells are at the highest level responsible for the direct killing of tumor cells through the secretion of cell membrane perforating molecules, i.e., cathepsin C, perforin, granzymes. The second mechanism, induced by CD8+ Tcells, which leads to cell apoptosis, is their expression of the Fas ligand, which, after binding to its receptor on the targeted cell, induces caspases and endonucleases leading to DNA impairment [31]. Previously, it was shown that a low number of CD8+ Tcells correlates with enhanced tumor growth and poor prognosis [32,33]. Detailed research points to a zeta chain as a key factor mediating antitumor response. In the case of oral cancer and Hodgkin’s Disease, a low level of the zeta chain in TCR in peripheral Tcells is correlated with a more advanced stage of disease [34,35]. Moreover, enhancing TCR signaling by binding immunoglobulin superfamily member 4 (IGSF4) to the zeta chain significantly decreased tumor size and weight in murine models with implanted melanoma cells and reduces the occurrence of metastatic colonies. [36]. Still, the high activity of IDO1, and thus a decreased TRP level leads to abnormal activity of immunosuppressive Tregs, whose pro-tumorigenic activity is based on crosstalk with other immune cells, as well as on a direct impact on cancer cells. Those cells express CD73 and CD39 endonucleotidases, which take part in adenosine production [37]. The overexpression of the latter was connected to an increased number of occurring liver metastasis in colorectal cancer (CRC) in the murine model and correlates with the worst prognosis and a poor outcome in patients [38]. Adenosine, whose production from ATP and AMP is enhanced in the presence of Treg, affects tumor development by activating PI3K/Akt/mTOR pathway and upregulating metalloproteinases that stimulate invasiveness and migration capacity of malignant cells [39]. Additionally, the whole loop leading to adenosine excretion leads to angiogenesis, caused by an intensified production of VEGF [40]. Besides the direct act on processes promoting oncogenesis, an elevated number of Tregs, and thus IDO1 activity, leads to the development of the immunosuppressive tumor microenvironment. High activity of Tregs limits interleukin 2 (IL-2) production, with the following CD8+ lymphocyte inhibition [41]. Other cells from the immune system, whose activity is at least in part controlled by Tregs are cancer-associated fibroblasts (CAFs), macrophage type 2 (M2) cells, regulatory B cells (Bregs), and myeloid-derived suppressor cells [42]. It should be emphasized here that the latter can secrete IDO1 [43], which loops the course of events even more. All of them are considered to be tumor promoters, their activity being increased under the impact of Treg. The outcomes of the experiments confirm the crucial role of Treg in cancerogenesis. In a few cancer types, i.e., gastric, breast, renal a higher intratumoral Treg level correlates with a worse prognosis [44,45,46]. Additionally, it was reported that high Treg activity contributes to chemoresistance [47,48]. On the other hand, clinical reports indicate the inhibition of inflammatory response by Treg, which in a further perspective leads to a decreased level of occurring tumor [49]. Taking into account the double role of Treg in cancer development, and IDO1 effect on both subpopulations of lymphocytes—the downregulation of CD8+ cytotoxic Tcells and the upregulation of immunosuppressive Treg it seems reasonable to focus on the ratio between these lines as the most proper prognostic factor in the context of IDO1 activity [50]. The outcomes from oncological patient samples confirm that a high CD8+/Treg ratio, and hence a low IDO1 activity, is associated with a more favorable prognosis [51,52,53]. The described effects are the results obtained by the activation of GCN-2 kinase. However, it was mentioned before that TRP depletion affects mTOR kinase, with its inhibition. This results in a decreased number of cytotoxic and helper T cells and an increased number of immunosuppressive Treg in the general population. Besides an indirect impact of IDO1 on cancer cells, it excretes its own direct effect. In bladder cancer cells, IDO1 inhibition leads to a limited colony formation, an increased E-cadherin expression with a concomitantly reduced N-cadherin and vimentin presence. This in total points to IDO1 being a promoter of epithelial-to-mesenchymal transition and a factor that facilitates a gain in motility capacity by cancer cells. Moreover, in the same experiment, the authors show a reduced ability for tubule formation by HVUECs and thus decreased angiogenesis after silencing IDO1 [54]. Those results are in the line with the results obtained by Pan et al., who reported decreased cell invasiveness and migratory ability in lung cancer cells after IDO1 silencing [55]. The same authors discovered that IDO1 presence conditions sufficient vessel density and the progression of vessel mimicry, which in short is unnecessary for the proper angiogenesis process. Another trial focused on lung cancer cells confirms that IDO1 activity is crucial for metastasis occurrence, and its inhibition improves patient outcomes [56]. The described data point to mutual permeating of immunological processes and cancer development, as well as indicate IDO1 as a significant factor that takes part in this mechanism.

## 5. TDO2, IDO2 and Their Role in Cancer Development

Two other rate-limiting enzymes are less studied than IDO1. Nevertheless, it is known that they are as well involved in maintaining oncogenesis. Their effect can partially be explained by similar to IDO1 activity toward TRP metabolism with its depletion in the local environment and the accumulation of immunosuppressive metabolites. From the pharmacological perspective, these overlapping events provide a reason for cancer cell resistance to IDO1 inhibition, as its role is taken over by the other two enzymes. However, the available data indicate that the expression level of IDO1, IDO2 and TDO2 differs between cancer types and that the occurrence of each enzyme alone can be an independent prognosis factor. This knowledge suggests that, at least in part, TDO2 or IDO2 exert a tumor-promoting role through different mechanisms than those described in IDO1 activity. In metastatic uveal melanoma, TDO2 but not IDO1 was found to be expressed in cancer tissue, in a constitutive manner [57]. In the same cell line, TNF-∝ (tumor necrosis factor-alfa) was pointed as a factor, which upregulates TDO2 expression, but has no impact on the IDO1 level. Additionally, in the case of triple-negative breast cancer, TDO2 seems to play a major role in disease progression, surpassing the importance of IDO1 [58]. Hsu et al. showed that TDO2 is expressed in lung cancer-associated fibroblasts which after knockdown disturb DCs differentiation and response from Th2 [59]. The role of the latter in cancer progression is based mainly on interleukin secretion. Th2 in a tumor microenvironment is a great source of IL-4, IL-5 and IL-13. Both IL-4 and IL-13, when in excess, have been connected with more aggressive cancers, enhanced metastasis, proliferation, and tumor growth. Detailed studies point to multiple mechanisms which are regulated by those interleukins. Well-known tumor-promoting signaling pathways, such as ERK1/2, Akt, mTOR, and STAT6 are among those induced by the mentioned factors. Moreover, it has been proved that IL-4 induces an expression of antiapoptotic proteins, such as Bax, BCL-xl, xFLIP, and contributes to sustained cancer growth through the induction of expression of glucose transporter—GLUT1 [60,61,62,63,64,65]. The aspect which cannot be omitted in the context of Th2 derived interleukins in cancer development is their role in macrophage polarization. Both IL-4 and IL-13 are involved in the differentiation of macrophages into pro-tumorigenic M2 [66], which are classified as tumor-associated macrophages (TAMs) [67]. A high presence of TAMs in the tumor microenvironment is associated with a poor prognosis in different cancer types as NSCLC, pancreatic carcinoma, ovarian cancer or gastric cancer [68,69,70,71]. Considering their involvement in cancerogenesis, it is worth mentioning, that they play a role at each stage of the ongoing process [72]. By the secretion of inflammation-promoting factors, i.e., TNF, IFN-g, ROS (reactive oxygen species) contribute to the establishment of the mutagenic microenvironment and facilitate the development of damaged cells [73]. In human glioma, ovarian cancer and clear cell renal carcinoma M2 have been reported to activate STAT3 signaling with enhanced proliferation and sustained survival of tumor cells [74,75]. As a source of proangiogenic factors, such as VEGF-A, and VEGF-C M2 contribute to angio- and lymphangiogenesis, respectively [76]. Their presence in the tumor microenvironment is connected with changes in the expression of EMT markers, such as E-cadherin, β-catenin, vimentin, and snail [77], which points to their involvement in aggressive phenotype development. TAMs enable metastasis formation and increase tumor cell invasiveness, via upregulation of metalloproteinases expression, which was reported both in vivo and in patient samples [78,79,80]. Noteworthy is the fact, that M2 can secrete exosomes, which contain specific miRNA or molecules such as ApoE and integrins, that in the next steps activate migration-inducing signaling pathways [81,82,83]. Moreover, the presence of M2 in a tumor microenvironment is considered as one of the factors responsible for the occurrence of chemoresistance [84,85] and its targeting could improve the efficacy of treatment [86]. The role of TDO2 in tumor development is not restricted only to the modulation of immune cell activity. It has been shown, that this enzyme is involved in enhancing the survival of circulating tumor cells (CTCs), through the participation in the development of resistance to anoikis (a form of programmed cell death) [87]. Due to AhR involvement in this process, the whole pathway is described below.

The last known rate-limiting enzyme involved in TRP metabolism is IDO2. Its role in tumorigenesis is significantly less examined than IDO1, thus a lot of questions about its involvement in cancerogenesis remain. Nevertheless, an important contribution of both IDO1 and IDO2 in cancer development, not mentioned before, is their involvement in NAD^+^ production. This dinucleotide as a final product of TRP metabolism is used as a source of energy for maintaining cells’ functions and viability. In a tumor microenvironment, an increased level of NAD^+^ has been shown to exert an immunosuppressive effect by the inhibition of T cell survival, proliferation and cytotoxic activity [88]. Additionally, its further metabolism takes part in enhancing immune evasion by promoting PD-L1 expression in tumor cells [89]. Its upraised level is connected with a poor prognosis and decreased overall survival in the number of solid tumors, i.e., non–small cell lung cancer, renal cell cancer, ovarian cancer, gastric cancers [90]. This ligand works through the PD-1 receptor expressed on different cell types, including activated Tcells. After interaction, the signaling from TCR attenuates [91], which is described above in the context of maintained tumor development. Additionally, recent reports suggest that PD-L1 has a direct impact on cancer cells, therefore it intensifies tumorigenesis not only via the modulation of the immune response. The correlation between an elevated level of PD-L1 and enhanced EMT was reported in a few cancer types, i.e., head and neck squamous cell carcinoma, lung adenocarcinoma, gastric cancer [92,93,94]. Wang et al. showed that in renal cancer cells PD-L1 overexpression induces the expression of SREBP-1c, a factor involved in cell lipogenesis. The authors connected this event with EMT induction and intensified cancer cell migration [95]. Other immune-independent effects caused by PD-L1 include the activation of mTORC1 and Ras/ERK and signaling cascades, with the following sustained growth of melanoma and ovarian cancer cells and enhanced EMT in glioblastoma multiforme, respectively [96,97]. Moreover, Mandarano et al. showed a correlation between upregulated IDO2 and PD-L1 levels, which indicates the existence of a link between IDO2 activity and enhanced cancerogenesis in a PD-L1-dependent manner [98]. Besides affecting PD-L1 expression, NAD^+^ contributes to stem cell proliferation and pluripotency [99]. However, it still needs to be confirmed if this nucleotide plays a role in maintaining the viability of cancer stem cells. There still exists a high need to define other mechanisms of IDO2 via which it contributes to cancerogenesis. However, reports showing an upraised level of this enzyme in human cancer tissues suggest that it is an important factor, which cancer cells use to maintain their survival in a host organism.

## 6. AhR Agonists—Carcinogenesis Modulators

The abovementioned effects are not only the ones that allow us to categorize the described enzymes as tumor modulators. Their wide role is also based on the production of metabolites that are involved in oncogenesis. Kynurenine, 3-hydroxykynurenine and kynurenic acid are characterized by a common feature—aryl hydrocarbon receptor activation. AhR is a transcriptional factor, which after interacting with a ligand translocates from the cytoplasm to the nucleus and modulates there the expression of targeted genes. Its final effects are ligand-specific, but due to the topic of this manuscript, here, only the effects caused by TRP metabolites are described (Figure 2).

It was suspected that AhR is a pro-tumor agent, although evidence suggests that, depending on cancer type it may have an opposite, tumor-suppressive role. Here it needs to be emphasized that, like enzymes, AhR activation can have a direct effect on cancer cells or contributes to immunosuppressive microenvironment development. Kynurenine was shown to induce Akt and STAT3 signaling pathways in renal cancer cells, and increase their proliferation, migration and drug resistance [100]. Enhanced migration after Kyn/AhR interaction was observed in HCC (hepatocellular carcinoma) and the authors linked it with changes in the level of EMT markers, as well as the overexpression of MMP9 [101]. Another reported mechanism, which contributes to migration, is the expression of aquaporin 4 (AQP4) observed in glioma cells after Kyn/AhR interaction [102]. Due to the regulation of cell volume, the contribution to filopodia formation and changes in shape, AQP4 enhances cell motility and metastasis [103,104]. Additionally, the high activity of this water channel was correlated with increased VEGF expression and related angiogenesis, as well as upraised metalloproteinase-2 activity [105,106]. Moreover, circulating cells that are detached from the extracellular matrix are highly vulnerable to a specific cell death type called anoikis. D’Amato et al. showed that TDO2/Kyn/AhR axis contributes to gaining a resistance to anoikis by cancer cells and by these means promotes migrating cells survival and facilitates metastasis formation (Figure 3) [87].

Kyn contributes to the development of resistance to classical apoptosis as well and that was reported in breast cancer cells [107]. Moreover, this metabolite in an AhR-depended mechanism causes cycle cell arrest in G0/G1 phase via the overexpression of cell-cycle inhibitor p27 in tumor repopulating cells (TRCs) [108]. This leads to cell dormancy—a state in which cells poorly proliferate and are highly resistant to pharmacological intervention (Figure 1). Shi et al. showed that blocking IDO1/Kyn/AhR pathway restores a cell cycle and induced apoptosis of TRCs [109], thus may be a proper strategy in disease relapse prevention. Additionally, TRCs release Kyn which then upregulates PD-1 expression on CD8+ Tcells by AhR activation (Figure 4) [110]. Li et al. showed that Kyn activates AhR expressed on DCs, and this interaction results in a switch from an immunogenic to tolerogenic subtype [111]. Tolerogenic DCs contribute to inducing Tcell differentiation into immunosuppressive Treg. Mezrch et al. confirmed that the Kyn/AhR axis is involved in Treg production by interaction with DCs [112].

Another metabolite—kynurenic acid—is also involved in increased cancer development, by immunomodulation. DiNatale at al. reported that KYNA treated MCF-7 cells overexpress IL-6, and this effect largely depends on AhR activation [113]. It was previously shown, that IL-6 expressing MCF-7 are characterized by the EMT phenotype and an increased metastatic potential [114,115,116], so in linking these two reports it is proper to consider KYNA/AhR interaction as a factor enhancing tumor invasiveness. The gathered data link AhR activation with ongoing cancerogenesis, concomitant inflammatory response and an immunosuppressive microenvironment. Nonetheless, in neuroblastoma Kyn via AhR activation lead to KISS1 gene overexpression and in this mechanism it was shown to decrease tumor metastasis and improve patient overall survival [117]. This is in the line with a few other reports, which shows that AhR acts like a tumor suppressor in liver cancer, colorectal cancer, melanoma, and prostate cancer [118,119,120,121]. Although, this effect was obtained in a KP-independent mechanism, further research is still needed to reveal if any of TRP metabolites have a tumor suppressive role through AhR activation.

## 7. Other Carcinogenesis Modulating Effects of TRP Metabolites

Besides the fact that the effects exerted in mechanisms depend on AhR activation, individual metabolites are also involved in cancerogenesis in different ways (Figure 4). Experimental evidence points to a correlation between an upraised level of circulating TRP derivatives and a higher risk of cancer overall or a more advanced grade of the disease [122,123]. However, their exact role has not been well studied yet the contribution of TRP metabolites to cancerogenesis is burdened with high ambiguities and controversies. They are connected with an undiscovered character of some metabolites, which seems to be pro and anti-cancerous at the same time. Due to this fact, whole KP remains a great subject for further studies and the establishment of its role will help to understand the pathogenesis of cancer. TRP derivatives are considered immunomodulators, but some of them seem to directly affect cancer cells. This paragraph gathers known data about TRP metabolites, tries to explain their complex role in cancerogenesis, and illustrates gaps and needs which should be included in further examinations. 3-hydroxykynurenine and 3-hydroxyanthranilic acid are considered DNA damaging carcinogens. This effect is exerted by the production of H_2_O_2_ affected by DNA single-strand breaks [124]. This leads to genome instability, which is a well-known feature of cancer cells. Other metabolites, i.e., kynurenine, quinolinic acid and cinnarbinic acid—a direct metabolite of 3HAA—are as well considered as ROS generators [125], which further impair different structures leading to enhanced tumor development. Song et al. showed that Kyn-derived ROS induces caspase release in NK (natural killers) cells and leads to their apoptosis [126]. NK cells are elements of an innate immune response, which are responsible for maintaining an anti-tumor response. Through direct cytotoxic activity achieved by the secretion of perforines and grazymes and/or by the induction of Fas-depended apoptosis, they eliminate malignant cells from the host organism [127]. Their depletion in the tumor microenvironment is one of the mechanisms which cancer cells use to avoid immunosurveillance. In a murine model of NSCLC, NK depletion was correlated with spontaneous lung metastasis achieved without affecting primary tumor growth [128]. A similar conclusion comes from tests conducted on colorectal mouse xenografts treated with tofacitinib-NK-reducing compound. In those animals, increased lung metastasis after the treatment was observed [129]. Additionally, Aydin et al. showed that the pro-metastatic effect exerted by a reduced number of NK is achieved in a ROS-depended mechanism [130], which lends a weight of argument that ROS-generating TRP metabolites enhance metastasis. Another effect leading to oxidative stress induction is mitochondrial dysfunction. In this context, QUIN was pointed as a harmful compound [131]. Mitochondrial failure has been connected with high aggressiveness, ongoing EMT and increased metastatic potential and chemoresistance of cancer cells [132]. Due to this, QUIN may be suspected to modulate these processes, but similarly to the abovementioned derivatives, its role is yet to be discovered. An elevated level of ROS is connected with all stages of tumor development, but detailed information about the contribution of TRP metabolites in specific processes has not been established as yet and this leads to a necessity to develop research in this direction.

Another ROS-independent tumor-promoting effect was reported by Bishnupuri and colleagues [133]. They show, that in colon cancer cells Kyn and QUIN activate PI3K/AKT/GSK3β which results in β-catenin translocation from the cytoplasm to the nucleus. In lung cancer cells kynurenine through Akt phosphorylation activates two other pathways—cAMP response element-binding protein (CREB) and with-no-lysine (K) protein kinases (WNK)—which were connected with the enhanced spheroid formation and migration, respectively [134]. Phosphorylation of a CREB signaling was previously connected with an enhanced expression of metalloproteinases 2 and 9, filopodia formation and an increased level of EMT markers [59,135]. On the other hand, the WNK kinases family affects downstream factors, including TGF-β or NF-kB (nuclear factor kappa B), known as tumor promotors [136]. Kynurenine acid ) has been reported as an G protein-coupled receptor 35 (GPR35) agonist (Figure 5) [137,138].

This receptor was found to be overexpressed in breast cancer, lymph nodes of colon cancer patients, which is connected with higher tumor aggressiveness and poor prognosis [139,140]. Another evidence of GPR35 involvement in migration comes from the trial lead on intestinal epithelial cells intentionally damaged by chemotherapy. Wang et al. showed decreased cell migration after GPR35 inhibition [141]. GPR35 is also expressed on macrophages, and activated under KYNA treatment [142]. Pagano et al. reported that the activation of GPR35 in those cells is connected with enhanced neoangiogenesis, tumor tissue remodeling and enhanced tumor growth in murine colon cancer xenograft [143]. This outcome suggests that KYNA by affecting macrophages may have an impact on colon cancer cells. Its contribution to other tumor types still needs to be confirmed. The function of macrophages is also impaired by 3-HAA (Figure 5). This TRP metabolite inhibits PI3k/AKT and mTOR signaling pathways, which leads to inhibition of NF-kB activation and attenuates nitric oxide (NO) production. Moreover, it also enhances the differentiation of macrophages into the M2 phenotype [144], whose role in cancerogenesis is described above. Oh et al. also showed decreased NO production by 3-HAA stimulated macrophages. Detailed research on possible mechanisms revealed that 3-HAA enhances HO-1 expression, with the following inducible nitric oxide synthase (iNOS) inhibition and a decrease in NO production. Although the role of M2 in carcinogenesis is clear, the level of NO and macrophage iNOS activity is not so clear-cut and the role of 3-HAA should be further explored. Besides macrophages, 3-HAA as a strong immunomodulator affects other immune cells (Figure 6). In dendritic cells, it attenuates the production of IL-6, IL-12 and TNF-∝ and inhibits their maturation and activation by interrupting p-JNK and p-p38 signals. The authors also pointed to a decreased ability of DCs for stimulation of Tcell activation, proliferation and differentiation under the influence of 3-HAA (Figure 5) [145]. Moreover, 3-HAA directly affects Tcells. Piscianz et al. proved that 3-HAA induces cell death in activated Tcell subpopulation [146], which is a well-known mechanism used by cancer cells in immune escape [147]. Lee et al. showed selective Th1 but not Th2 apoptosis induced by 3-HAA and QUIN. This discovered mechanism depends on cytochrome c release and caspase activation as independent from Fas/Fas ligand interaction [148]. Moreover, 3-HAA alone eliminates already activated Tcells, by decreasing intracellular glutathione levels [149]. Altogether, 3-HAA leads to an increase of Th2 subpopulation, whose procancerous activity is described above. As mentioned at the beginning 3-HKYN and 3-HAA also induce strong apoptosis of CD4+ Tcells with a simultaneous increase of immunosuppressive CD4+ FoxP3 Tcell subpopulation [150]. These observations additionally enhance the pro-tumorigenic role of TRP metabolites.

Nonetheless, as mentioned before, KP contribution to cancerogenesis is not clear. Walczak et al. gathered information about KYNA level in different cancers, and showed that depending on the type of disease, the level of this metabolite is decreased or upraised, but the mechanism of this phenomenon is not discovered [151]. However, this suggests that in some conditions, KYNA may serve as a tumor suppressor (Figure 7).

In HT-29 colon cancer cells, this derivative was shown to inhibit p21 Waf1/Cip1 with following DNA synthesis inhibition and decreased proliferation [152]. The same mechanism was reported for renal cancer cells [153]. Moreover, the authors point to decreased migration of malignant cells after KYNA stimulation, which stays in a line with another experiment conducted on glioma cells. In these brain tumor cells described metabolite was pointed as a factor antagonizing glutamate activity, which resulted in DNA synthesis inhibition and reduced migration [154]. KYNA is also considered as an alpha-7 nicotinic acetylcholine receptor (a7nAChR) inhibitor [155]. This interaction mainly matters in neuronal disorder, nevertheless, recent reports emphasized the role of nicotinic acetylcholine receptors in oncogenesis. The blockage of a7nAChR reversed acetylcholine-induced cell migration and invasiveness in NSCLC cells and gastric cancer cells via blocking signaling by MEK/ERK pathway [156,157]. Moreover, in gastric cancer, cholangiocarcinoma, this receptor inhibition was connected with reduced expression of EMT markers [158]. Scientific reports also indicated increased apoptosis of breast cancer and cholangiocarcinoma cells after a7nAChR silencing [159]. KYNA is also considered a tumor suppressor due to the inhibition of fibroblast growth factor-1 (FGF-1) release [160], which is involved in MMPs activation, angiogenesis, tumor progression, promotion of cancer cells stemness and is associated with worst patients’ prognosis [161,162,163,164,165,166]. The blockage of the FGF-1 signaling pathway causes a cell cycle arrest in phase G0/G1, thus decreased cancer cell proliferation. It also reduces tumor growth and occurring metastasis [167,168]. Another metabolite described before as a tumor promotor 3-HAA also seems to be a double-edged compound. In the human umbilical vein endothelial cells (HUVECs), which serve as a model for angiogenesis evaluation 3-HAA inhibits monocyte chemoattractant protein-1 (MCP-1) secretion and vascular cell adhesion molecule-1 (VCAM-1) expression [169]. These molecules belong to factors enhancing angiogenesis, cancer cell survival, invasiveness and metastasis in different tumor types [170,171,172,173]. VCAM-1 also facilitates circulating tumor cell adhesion to the vascular endothelium, and thus burrowing circulating cells in new organs [174]. Targeting these proteins and decreasing their level may help to limit metastasis, cancer progression and improve patient outcomes, thus 3-HAA by this mechanism may work as a tumor suppressor. Another TRP metabolite, with potential antitumor activity, is picolinic acid In the murine model Ehrlich ascites carcinoma treatment with PA significantly reduced tumor growth and improved longevity [175]. A similar result was obtained by Ruffman et al. who observed a therapeutic effect of PA in mice bearing MBL-2 lymphoma cells [176]. Apart from these old experiments, new ones are not available, and mechanisms through which PA exerts its antitumor effect are not well known. Gathered together this information illustrate that KP remains a puzzle for scientists focusing on cancer pathogenesis. However, the establishment of its exact role may help to develop an effective therapy for patients in the future.

## 8. Compounds Targeting Kynurenine Pathway in Clinical Trials

Nowadays, even without full knowledge about TRP metabolites in tumorigenesis, efforts are taken to develop compounds, which by KP modulation will serve as anticancer drugs. Due to the multistep process, the targeting of different parts of this pathway is being considered. Unfortunately, promising preclinical results more than once disappointed at the clinical stage. It may be connected with overlapping events and a huge number of factors contributing to TRP metabolism, and their ambiguous character. Due to that further studies need to be taken to reveal which pathways should be silenced, and which enhanced to ensure an effective therapy based on KP modulators. Perhaps a combination therapy with compounds targeting other metabolic pathways will be needed, and indeed recent clinical trials in huge majority examine a combination therapy. This paragraph summarizes, in short, KP modulators, which are under development as an anticancer therapy. One of the approaches focuses on IDO1, IDO2 and TDO2 inhibition, due to their pro-cancerogenous activity. Inhibitors are divided by their affinity to each enzyme, and thus from available drugs selective IDO1 or TDO2 inhibitors or a non-selective IDO1/TDO2 inhibitor can be distinguished. Epacadostat—a selective IDO1 inhibitor—is examined at the highest level [177,178]. Since monotherapy with epacadostat was not more favorable when compared to tamoxifen in ovarian cancer [179], the connection with other compounds has been tested. Phase III of the trial checking the efficacy of epacadostat with pembrolizumab versus pembrolizumab alone in melanoma showed no predominance in the drug combination [180]. A similar conclusion was reported by Kelly et al. who checked a combination of epacadostat and pembrolizumab in patients with advanced sarcoma [181]. A clinical trial evaluating the combination with another monoclonal antibody, atezolizumab, toward NSCLC also did not show satisfying results [182]. Those surprisingly disappointing results impeded research on IDO1 selective inhibitors, nevertheless, some novel compounds are tested in new combinations and indications. The effectiveness of one of them, BMS-986205 combined with biological compounds in endometrial cancer or endometrial carcinosarcoma, liver cancer, bladder cancer, melanoma, NSCLC, oral cavity squamous cell carcinoma, glioblastoma, kidney cancer and gastric cancer will be tested in clinical trials, which are now in the recruiting phase [183,184,185,186,187,188,189,190]. The trial comparing BMS-986205 with nivolumab to nivolumab alone efficacy is terminated, but the results are not available yet [191]. KHK2455 and LY3381916 are both in early development. The application of the first one in bladder cancer in combination with anti-PD-L1 therapy will be checked in a trial that is now in the phase of recruiting patients [192]. The second compound, also with anti-PD-L1 coadministration in different types of solid tumors was tasted in a terminated trial, the results of which are not available yet [193]. Due to the newly recognized role of TDO2 in cancer progression, molecules inhibiting its activity are also under development. There are not a lot of data available in this field, however, one selective TDO2 inhibitor—680C9—is in a preclinical test and it sensitizes glioblastoma cells on genotoxic treatment when combination therapy is applied [194]. Other selective TDO2 inhibitors are in early development and further investigation is highly needed [195,196]. The lack of the expected efficacy of selective inhibitors forced to raise a question about the cause of this effect. One hypothesis is based on the possibility of taking over the role of the inhibited enzyme by another, still an active one. Due to that fact, the compounds with inhibitor activity towards two enzymes are under enhanced development and include indoximod (1-MT), its prodrug NLG-802 and navoximod [197,198,199]. The clinical significance of 1-MT in different solid tumors and in acute myeloid leukemia was evaluated alone or in combination therapy [200,201,202]. Attempts have been made to co-administrate this compound with classical chemotherapy, such as nab-paclitaxel or gemcitabin for pancreatic cancer, temozolomid for brain cancer taxanes in breast cancer or docetaxel in solid tumors [203,204,205,206]. However, despite good tolerance, there is a lack of predominance of combination therapy [205]. Still, indoximod was reported to increase the effectiveness of DNA damaging chemotherapy, without changes in its toxicity [202]. Additionally, it was also showed that 1-MT improves the efficacy of radiotherapy treatment [207]. NLG-802 was evaluated only in Phase I for solid tumors, however, the results of this trial are not available. Navoximod was tested in Phase I of the clinical trial combined with ateolizumab. The trial was conducted on patients with solid tumors generally. As before, a good toxicity profile was illustrated, however no benefits from the combination therapy were shown [208]. In turn, the outcomes from Phase Ib indicated that the monotherapy of recurrent advanced solid tumors resulted in stable disease response [209]. Other IDO/TDO2 inhibitors, such as HTI-1090, DN1406131 are in early development, and no final results are available yet [210,211]. Compounds such as RG70099, EPL-1410, CB548, CMG017 are in preclinical tests and show potential antitumor activity [212,213,214]. MK-7162 another compound targeting TRP catabolizing enzymes has no discovered mechanism, nevertheless, it is included in clinical trials for adult patients with solid tumors in a combination with pembrolizumab [215]. As of now, the results from this phase are not known. Another approach targeting KP enzymes is based on the occurrence of specific cytotoxic Tcells directed toward IDO1, IDO2, TDO2. Detailed studies lead to a discovery, that this Tcell subpopulation has a direct cytotoxic activity toward tumor cells and immunosuppressive DCs, Th2 [216,217,218]. For this reason, enzyme-derived epitopes are used as a vaccine to enhance response from effector Tcells. This therapeutic option is already in a clinical trial in NSCLC and melanoma in combination with temozolomide, ipilimumab, epacadostat [216,219,220,221]. The available results point to good safety profile, however a bigger group of patients is needed to confirm its efficacy. The next structure responsible for pro-tumoral effect of TRP metabolites is AhR and its inhibition may serve as a target in cancer treatment. In a preclinical test on mice xenografts with oral, breast and skin cancer HP163 reduces both tumor growth and a number of immunosuppressive cells [222]. Another small molecule AhR inhibitor—BAY218, used in the murine model—reduced tumor growth and enhanced immune response by increasing the infiltration of CD8+ T and NK cells with simultaneous reduction with regard to suppressive GR1+ myeloid cells and M2 macrophages [223]. A significant activity toward M2 macrophages under AhR inhibitor treatment was also observed by Garcia et al. [224]. They show in in vitro study that IDE-AhRi-1 administration leads to the full arrest of M2 suppressive effect and the enhancement of Tcell activity. The first clinical trial with the use of AhR inhibitor is in the phase of recruiting patients, hence any clinical significance of this strategy will be known in the future [225].

As observed, the whole KP is a very potent therapeutic target, however high hopes connected with IDO1 or TDO2 inhibition up to date turn out to be promising in preclinical tests and not highly effective in clinical practice. It is possible that the described complexity of KP and the dual role of metabolites in cancerogenesis is a reason for this failure. It shows how important it is to focus on further and detailed research on each TRP metabolite to find a golden mean. Positive results of the preclinical test (although disappointing in further steps) suggest that targeting KP is a potent strategy for the establishment of alternative treatment and improvement of needs. Moreover, in clinical practice, KP has one more application. Besides serving as a target for therapy, measurement of the level of the circulating metabolites or Kyn/TRP ratio is used as a biomarker of occurring cancer disease. Determination of these quantities in the blood or urine derived from patient samples has been previously used to define the advancement of the disease and helped to predict patient prognosis in the case of few cancer types, i.e., renal cell carcinoma, bladder cancer, breast cancer, colon cancer, prostate cancer or gastroesophageal cancer [226,227,228,229,230,231]. This additionally emphasizes the usefulness of the practical application of the kynurenine pathway in oncological diagnosis and treatment.

## 9. Other Aspects of TRP Metabolism

As mentioned at the beginning of this manuscript, KP is not the only pathway through which TRP is metabolized. The importance of the serotonin pathway in the context of enhancing oncogenesis has also increased significantly in recent years. The available data indicate an increased growth of cancer cells under the influence of serotonin in prostate cancer, breast cancer, small-cell lung cancer, colorectal cancer, cholangiocarcinoma, hepatocellular carcinoma, and glioma [232]. Moreover, the activity of this monoamine has been associated with an intensified epithelial–mesenchymal transformation process, increased cell migration abilities, and thus an increase in metastatic potential [233]. In vascular endothelial cells, 5-HT activates angiogenic-signaling kinases, which confirms its involvement in the vascularization of growing cancerous tumors [234,235]. In addition, the discussed monoamine is also classified as an immunomodulator, however, its pro and anti-tumor role are yet to be established. By affecting macrophages, it contributes to the formation of the immunosuppressive M2 population, on the other hand, 5-HT increases the cytotoxicity of natural killers, contributing to the increased eradication of cancer cells [236]. The studies also showed increased proliferation of B lymphocytes under the influence of 5-HT, however, their pro or antitumor effect exerted in the tumor microenvironment is unclear [237,238]. A similar effect was observed on Tcell proliferation after 5-HT stimulation [236]. The involvement of the serotonin pathway in cancerogenesis is another extensive topic, and cannot be described in the given manuscript in detail. Nevertheless, it is important to keep in mind that the role of TRP catabolism in cancer development is not restricted to a kynurenine pathway.

## 10. Conclusions

The above information shows how extensive and complex the role of tryptophan metabolism in modulating oncogenesis is. The world of science is facing an enormous challenge that must be resolved in order to understand the role of this metabolic pathway in cancer development. Despite the growing number of sources of information about individuals’ role and their involvement in cancerogenesis, a lot of them are still a puzzle and give an opportunity for further significant research. Overlapping immune mechanisms and a classic well-known process of tumorigenesis additionally emphasize a need for exploring the role of immunomodulators as tumor promotors or suppressors. Gaining this knowledge may change the face of oncological treatment in the future and improve patients’ survival and quality of life.

## Figures and Tables

**Figure 1 cancers-13-02667-f001:**
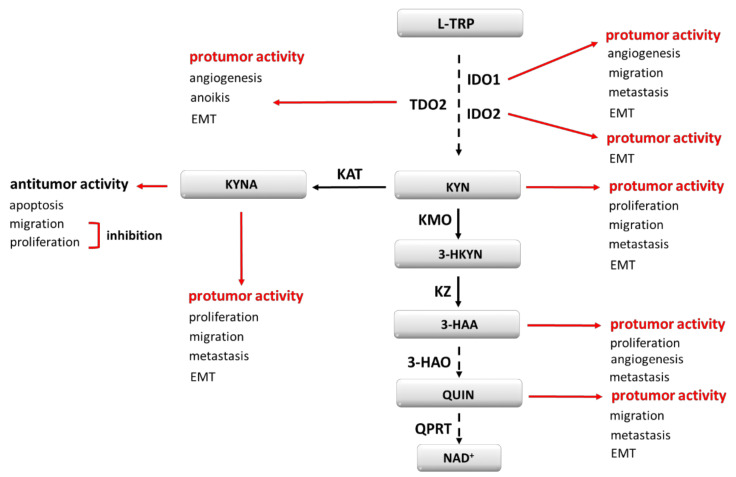
Kynurenine pathway elements as cancerogenesis modulators. EMT—epithelial-to-mesenchymal transition; 3-HAA—3-hydroxyanthranilic acid; 3-HAO—3-hydroxyanthranilate-3,4-dioxygenase; 3-HKYN—3-hydroxykynurenine; IDO1, IDO2—indoleamine oxidase 1, 2; KAT—kynurenine aminotransferase; KMO—kynurenine 3-monooxygenase; KYNA—kynurenine acid; KYN—kynurenine; KZ—kynureninase; L-TRP—tryptophan; NAD^+^—nicotinamide adenine dinucleotide; QPRT—quinolinic acid phosphoribosyltransferase; QUIN—quinolinic acid; TDO2–tryptophan 2,3-oxidase.

**Figure 2 cancers-13-02667-f002:**
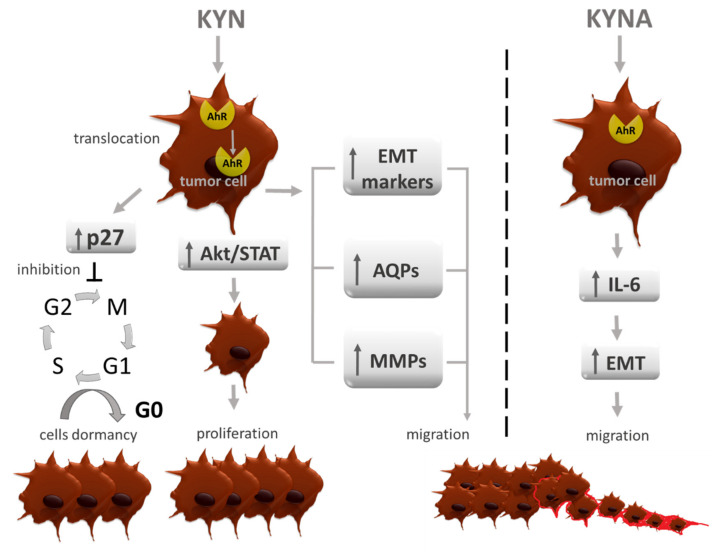
AhR—dependent pro-tumorigenic activity of TRP metabolites. AhR—aryl hydrocarbon receptor; Akt—protein kinase B; AQP4—aquaporin 4; EMT—epithelial-to-mesenchymal transition; IL-6—interleukin 6; KYN—kynurenine; KYNA—kynurenic acid; p27-tumor suppressor; MMPs—metalloproteinases; STAT3—signal transducer and activator of transcription 3.

**Figure 3 cancers-13-02667-f003:**
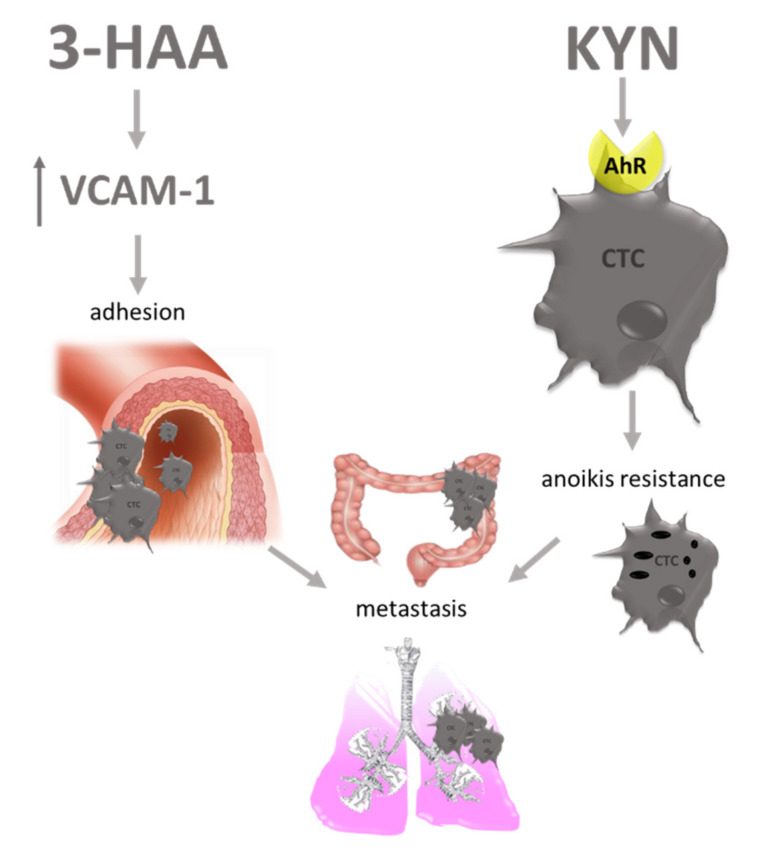
Effects of TRP metabolites on circulating tumor cells (CTCs). 3-HAA—3-hydroxyanthranilic acid; AhR—aryl hydrocarbon receptor; CTC—circulating tumor cell; KYN—kynurenine; VCAM-1—vascular cell adhesion molecule-1.

**Figure 4 cancers-13-02667-f004:**
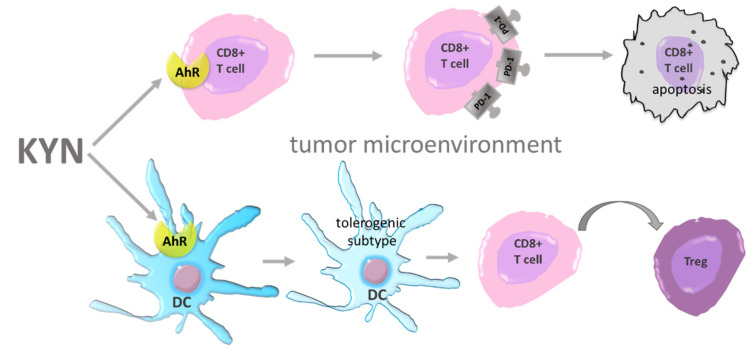
Immune evasion—impact of KYN on the tumor microenvironment. AhR—aryl hydrocarbon receptor; CD8^+^ T cell—lymphocyte T CD8^+^; DC—dendritic cell; KYN—kynurenine; PD-1—programmed death receptor 1; Treg—regulatory Tcells.

**Figure 5 cancers-13-02667-f005:**
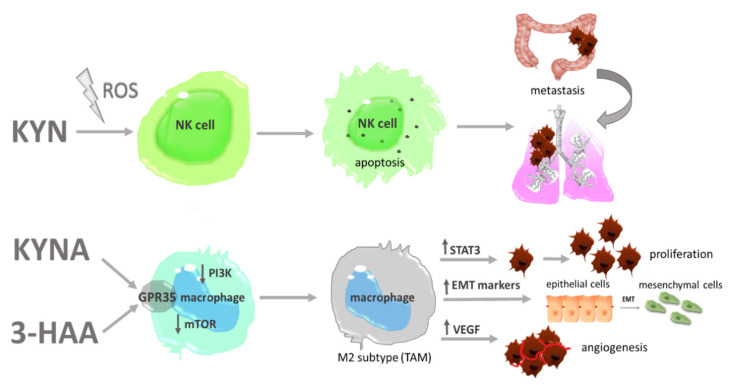
AhR—independent pro-tumorigenic activity of TRP metabolites. 3-HAA—3-hydroxyanthranilic acid; EMT—epithelial-to-mesenchymal transition; GPR35—G protein-coupled receptor 35; KYN—kynurenine; KYNA—kynurenic acid; mTOR—mechanistic target of rapamycin; NK cell—natural killer; PI3K—phosphatidylinositol 3 kinase; ROS—reactive oxygen species; STAT3—signal transducer and activator of transcription 3; TAMs—tumor associated macrophages; VEGF—vascular endothelial growth factor.

**Figure 6 cancers-13-02667-f006:**
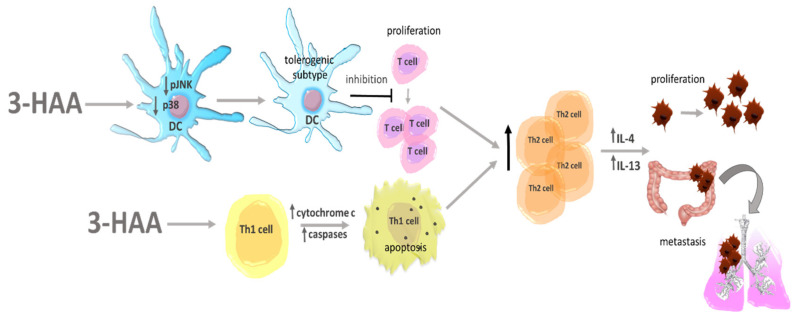
Impact of 3-HAA on the tumor microenvironment. 3-HAA—3-hydroxyanthranilic acid; DCs—dendritic cells; IL-4, 13—interleukins 4, 13; p38—phosphorylates specific transcription factor; pJKN—c-Jun N-terminal kinase; T cell—lymphocyte TTh—helpers lymphocytes T.

**Figure 7 cancers-13-02667-f007:**
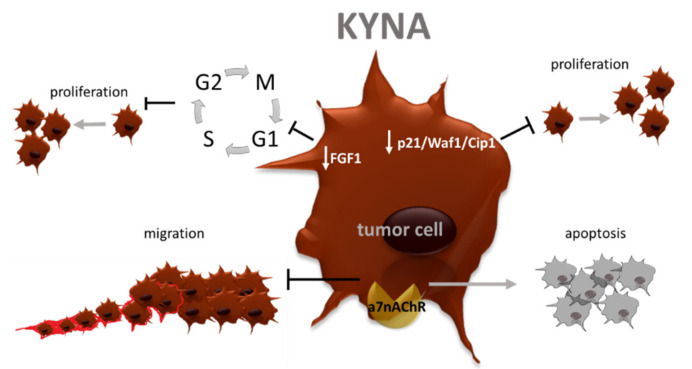
Antitumor effects of KYNA. a7nAChR—alpha-7 nicotinic receptor; FGF-1- fibroblast growth factor-1; KYNA—kynurenic acid; p21 Waf1/Cip1—cyclin-dependent kinase inhibitor 1.

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
