# Peer review of "Not Only Immune Escape—The Confusing Role of the TRP Metabolic Pathway in Carcinogenesis"

_cancers, 2021, doi:10.3390/cancers13112667_

Round 1

Reviewer 1 Report

This review paper summarized the role of tryptophan metabolism pathway in carcinogenesis. The topic is very interesting. However, the paper was not well written, there should be a logical flow. To providing a schematic diagram of general role of tryptophan metabolism pathway in carcinogenesis would help readers to have a whole picture on this topic.

Minor issue:
Some font size in the figures are too small.

Reviewer 2 Report

In this paper Authors have comprehensively described the involvement of tryptophan metabolism via the kynurenine pathway in carcinogenesis, thus providing several promising suggestions of therapeutic targets. Generally, the study is interesting, and the topic is timely because investigations of the TRP metabolic pathway underlying immunomodulators response would be helpful to develop targeted therapies. The paper is well done, although some descriptions should be less dispersive and always limited to the purpose of the manuscript. Of course, there would be a lot to say for each topic treated in the discussion but the authors should avoid long descriptions of some of the different actors who are involved in the discussion also to make the reading of the text more fluent and understandable. Therefore, in my opinion, it would be reasonable to focus the topics better by reducing where possible the discussion of the data. Here some other suggestion:

  • Authors should check abbreviations carefully. In fact, in some cases the abbreviation is used directly without indicating the complete definition, in other cases the definition is indicated after the first abbreviation use, other times never. Check for instance TRP, Kyn, KP, AhR, DCs and so on.
  • Line 175: “patents”, maybe is “patients”
  • Paragraph “4. TDO2, IDO2 and their role in cancer development”: this part, represent an example where the description of ILs secretion is too long and dispersive. Authors should try to abbreviate and focus on main topic of paper particularly by choosing the salient points linked to KP.

Reviewer 3 Report

I read the manuscript entitled “Not only immune escape - the confusing role of the TRP metabolic pathway in carcinogenesis”.

This paper disserts on evidence that some mechanisms that lead to immune escape by cancer cells involved tryptophan (TRP) metabolism through the kynurenine pathway (KP).

In particular available data about the kynurenine pathway and its modulating effects on disease development, through the impact on immune and cancer cells have been described.

The aim and objectives of the paper are interesting and the amount of citations are relevant.

However, some criticism persists:

  1. In order to characterize under different conditions changes in metabolism, the assay of different compounds (i.e. urinary biomarkers) should be considered.
  2. As a general observation dying cells established local conditions that suppress antitumour immunity to promote tumorigenesis. This is not kynurenine-exclusive effect, the authors should be able to give consideration to others amino-acid pathway.
  3. Considering the regulation of enzymes of proposed pathways, the possible changes of serotonin pathway (i.e. as pharmacological target) should be also evaluated.
